# Utility of FIB4-T as a Prognostic Factor for Hepatocellular Carcinoma

**DOI:** 10.3390/cancers11020203

**Published:** 2019-02-10

**Authors:** Kazuya Kariyama, Kazuhiro Nouso, Hidenori Toyoda, Toshifumi Tada, Atsushi Hiraoka, Kunihiko Tsuji, Ei Itobayashi, Toru Ishikawa, Akiko Wakuta, Ayano Oonishi, Takashi Kumada, Masatoshi Kudo

**Affiliations:** 1Department of Gastroenterology and Liver Disease Center, Okayama City Hospital, Okayama 700-8557, Japan; kariyama777@gmail.com (K.K.); kazunouso@gmail.com (K.N.); akikowakuta@gmail.com (A.W.); butanishinjuflower@gmail.com (A.O.); 2Department of Gastroenterology and Hepatology, Ogaki Municipal Hospital, Gifu 503-0864, Japan; tkumada@he.mirai.ne.jp (H.T.); tadat0627@gmail.com (T.T.); takashi.kumada@gmail.com (T.K.); 3Gastroenterology Center, Ehime Prefectural Central Hospital, Ehime 790-0024, Japan; hirage@gmail.com; 4Center of Gastroenterology, Teine Keijinkai Hospital, Sapporo 006-0811, Japan; ktsuj@keijinkai.or.jp; 5Department of Gastroenterology, Asahi General Hospital, Asahi 241-0801, Japan; itobayas@crocus.ocn.ne.jp; 6Department of Gastroenterology, Saiseikai Niigata Daini Hospital, Niigata 950-1104, Japan; toruishi@ngt.saiseikai.or.jp; 7Department of Gastroenterology and Hepatology, Kindai University Faculty of Medicine, Osaka-Sayama 589-8511, Japan

**Keywords:** ALBI-T, FIB4 index, FIB4-T, JIS score

## Abstract

Background: Most integrated scores for predicting the prognosis of patients with hepatocellular carcinoma (HCC) comprise tumor progression factors and liver function variables. The FIB4 index is an indicator of hepatic fibrosis calculated on the basis of age, aspartate aminotransferase (AST) levels, alanine aminotransferase (ALT) levels, and platelet count, but it does not include variables directly related to liver function. We propose a new staging system, referred to as “FIB4-T,” comprising the FIB4 index as well as tumor progression factors, and examine its usefulness. Method: Subjects included 3800 cases of HCC registered in multiple research centers. We defined grades 1, 2, and 3 as a Fibrosis-4 (FIB4) index of <3.25, 3.26–6.70, and >6.70 as FIB4, respectively, and calculated the FIB4-T in the same manner in which the JIS (Japan Integrated Staging Score) scores and albumin-bilirubin tumor node metastasis (ALBI-T) were calculated. We compared the prognostic prediction ability of FIB4-T with that of the JIS score and ALBI-T. Results: Mean observation period was 37 months. The 5-year survival rates (%) of JIS score (0/1/2/3/4/5), ALBI-T (0/1/2/3/4/5) and FIB4-T (0/1/2/3/4/5) were 74/60/36/16/0, 82/66/45/22/5/0 and 88/75/65/58/32/10, respectively. Comparisons of the Akaike information criteria among JIS scores, ALBI-T, and FIB4-T indicated that stratification using the FIB4-T system was comparable to those using ALBI-T and JIS score. The risk of mortality significantly increased (1.3–2.8 times/step) with an increase in FIB4-T, and clear stratification was possible regardless of the treatment. Conclusions: FIB4-T is useful in predicting the prognosis of patients with HCC from a new perspective.

## 1. Introduction

Liver cancer is the second leading cause of death due to cancer worldwide and was responsible for 788,000 deaths in 2015 [1]. The prognosis of hepatocellular carcinoma (HCC) is regulated by tumor factors and liver function of the background liver. Various integrated scores, such as the Okuda [2], Cancer of the Liver Italian Program (CLIP) [3], TOKYO [4], JIS [5], and albumin-bilirubin tumor node metastasis (ALBI-T) [6], incorporate both of these factors, as well as background liver factors representing synthetic and metabolic functions.

Liver fibrosis progresses from various factors, such as genetic factors, viral or non-viral inflammation, influence by drugs, is considered to be involved in carcinogenesis, is known to be a risk factor for HCC development following sustained virological response to hepatitis C virus (HCV), and can also determine the prognosis of patients with cirrhosis [7]. The FIB4 index was developed in 2006 by Sterling as a non-invasive method to diagnose liver fibrosis and combines aspartate aminotransferase (AST) levels, alanine aminotransferase (ALT) levels, platelet count, and age [8]. The index was originally used for staging liver fibrosis in HCV patients with HIV [8] and, thereafter, has been used to quantify fibrosis of various liver diseases, including HCV (Hepatitis C virus) or HBV (Hepatitis B virus) infection, alcoholic liver disease, and non-alcoholic fatty liver disease [9,10]. As the FIB4 index is calculated using only clinical laboratory test values and age, it is a completely objective evaluation, similar to the ALBI score. Moreover, the FIB4 index uses no numerical values directly correlated with liver synthetic ability or metabolic ability, such as albumin, total bilirubin, and prothrombin levels; therefore, not a single variable in this index overlaps with those in the ALBI and Child–Pugh scores. Another advantage of the FIB4 index is that it does not include the variables used in conventional integrated scores, which tend to vary with other factors.

In this article, we propose a new staging system for prognostic prediction in patients with HCC, called “FIB4-T,” which comprises the FIB4 index and several tumor factors, and we examine its usefulness.

## 2. Patients and Methods

### 2.1. Patients

Subjects used in this analysis included 3800 HCC cases registered in multiple collaborative hospitals (Ehime Prefectural Central Hospital, 1418 cases; Ogaki Municipal Hospital, 1248 cases; Asahi General Hospital, 799 cases; Okayama City Hospital, 193 cases; Saiseikai Niigata Daini Hospital, 142 cases) from 2000 to 2015. Diagnosis of HCC was via imaging modalities, including computed tomography (CT), magnetic resonance imaging (MRI), and angiography. The diagnostic criteria for HCC was based on previous reports of hyperattenuation at the arterial phase or hypoattenuation at the portal phase, determined using dynamic CT or MRI with tumor staining on angiography [11]. In cases with atypical findings we confirmed the diagnosis pathologically, using tissue obtained from a fine needle tumor biopsy. The 6th edition tumor, node, metastasis (TNM) staging for HCC was determined based on previous studies conducted by the Liver Cancer Study Group of Japan (LCSGJ) [12].

The study protocol conformed to the ethical guidelines of the World Medical Association Declaration of Helsinki and was approved by our institutional review board. The Ethics Committee is the IRB of Okayama City Hospital and the approval code was 29-43.

### 2.2. Construction of FIB4-T

Several reports have examined FIB4 grading in liver disease using different cut-off levels [13,14,15]. Since this study included subjects with HCC, many patients had advanced fibrosis and therefore we defined a FIB4 index of less than 3.25 as grade 1, which is a relatively high cut-off value previously proposed by Sumida et al. [13]. The rest of the patients were divided into two additional groups with either a FIB4 index of 3.26–6.70, defined as grade 2, or above 6.70, defined as grade 3. The FIB4-T score was created by simply adding the TNM stage (stage I, 0; stage II, 1; stage III, 2; and stage IV, 3) and FIB4 grade (grade1, 0; grade 2, 1; and grade 3, 2) (Table 1).

### 2.3. Evaluation of Scores

Two integrated scores, JIS [5] and ALBI-T [6], were used to evaluate the FIB4-T score. The JIS and ALBI-T scores are common integrated scores for predicting the prognosis of liver cancer and combine liver function and TNM staging. Survival curves of each score were estimated using the Kaplan–Meier method and compared using the log-rank test. The Akaike information criterion (AIC) of the integrated scores were compared to evaluate their discriminatory ability. The prognostic predictive power of each FIB4-T score was evaluated using the Cox proportional hazards regression model.

### 2.4. Statistics

Data was expressed as the median and range. Statistical analysis was performed using the log-rank test, Kaplan–Meier method, and the Cox proportional hazard model. A *p*-value < 0.05 was considered statistically significant. All statistical analysis was performed using Easy R (EZR; Saitama Medical Center, Jichi Medical University, Saitama, Japan) [16], a graphical user interface for R (The R Foundation for Statistical Computing, Vienna, Austria) [17].

## 3. Results

### 3.1. Patient Characteristics

The median age (range) at diagnosis was 70 (21–98) years old and 2779 cases (73.1%) were male. Two thirds of patients were positive for the HCV antibody, while percentages of patients positive for the Hepatitis B surface (HBs) antigen and patients without viruses were low (11.1% and 20.4%, respectively). Over 90% of patients were Child–Pugh A or B and more than half of patients (57.4%) had received surgical resection or local ablation therapies categorized as curative therapies (Table 2). The mean observation period was 37 months.

### 3.2. Patient Survival

Survival of patients was significantly different between FIB4 grades (*p* < 0.0001, Figure 1a) and between TNM stages (*p* < 0.0001, Figure 1b). Integrated FIB4-T scores, combining the FIB4 grade and TNM stage, clearly stratified the patients (*p* < 0.0001, Figure 2a–c). Distribution of FIB4-T in JIS score and ALBI-T were as shown in Figure 2d,e. The range of median survival time (MST) was widest for FIB4-T (0.2–9.8 years) compared to the FIB4 (3.0–7.0 years) and TNM stage (0.6–7.3 years) alone (Table 3). Regardless of treatment, stratification by FIB4T was possible (Figure 3). Survival curves of the patients stratified by treatment types (Surgery, RFA and TACE) in each FIB4-T grade were shown in Figure 4.

### 3.3. Stratification of FIB4-T Population

The same HCC patient cohort was stratified using three different integrated scores (FIB4-T, JIS, and ALBI-T) and compared (Figure 2). All scores were able to stratify the survival of the patients reasonably well, however, the difference between FIB4-T (0) and FIB4-T (1) was relatively small. The difference in MST between FIB4-T (0) and (1) was 20 months (*p* = 0.048), whereas the difference between JIS score (0) and (1) was 30 months (*p* < 0.0001), and the difference between ALBI-T (0) and (1) was 48 months (*p* < 0.0001). The discriminatory ability of FIB4-T was slightly lower than that of the ALBI-T or JIS score, according to AIC values of 26,460, 26,058, and 26,018, respectively (Table 4).

### 3.4. Risk of Mortality Based on FIB4-T

Based on an increase in the FIB4-T score, the risk of mortality increased 1.3–2.8 times/step. The difference became larger as the FIB4-T score increased (Table 5). The relative risk of a one-step increase in the FIB4-T score was lower than those of high alpha-fetoprotein, high des-gamma-carboxy prothrombin, and portal vein tumor thrombus; however, clear statistical differences were observed between adjacent FIB4-T scores.

## 4. Discussion

Several integrated scores comprised of tumor and background liver factors have been proposed as prognostic models of HCC. The first was the Okuda score [2], followed by the CLIP score [3]. Based on developments in screening methods and therapies for HCC, the JIS score [5] and ALBI-T grade [6] were previously proposed and are now routinely used. In this study, we proposed a new integrated scoring system, called FIB4-T, that incorporates the fibrosis index and several tumor factors. The score was calculated without any factors directly representing the synthetic and metabolic ability of the liver, such as albumin, total bilirubin, and prothrombin, which are used in existing integrated scores. Although the AIC value of FIB4-T, representing discriminatory ability, was slightly higher (worse) than the JIS score and ALBI-T, the stratification ability of FIB4-T was comparable. Moreover, the prognosis of patients was well stratified using the new factors of the FIB4-T score.

As an index of hepatic fibrosis, various validation studies have been done on FIB4 [8,9,13,14,15,18,19,20,21,22,23,24,25]. Usefulness of the index was demonstrated not only in patients with HCV but also in patients with non-alcoholic fatty liver disease and alcoholic liver disease. Liver fibrosis is known to be independent of, and correlates closely with, survival of the patients with chronic liver disease. In this study, we demonstrated the usefulness of an integrated score incorporating the FIB4 index, which is focused on the extent of fibrosis.

The prognosis of HCC is greatly influenced by tumor factors, and also importantly, by liver function. Therefore, serum albumin, total bilirubin, and prothrombin have been used as indicators of liver synthetic and metabolic ability in many integrated scores. However, albumin levels are easily influenced by dietary intake. Furthermore, increased bilirubin is frequently observed in cases of constitutional jaundice, such as Gilbert syndrome, and prothrombin is influenced by warfarin administration. Gilbert syndrome is prevalent in approximately 6% of the general population [26] and the use of anticoagulant, for preventing cardiovascular disease, is increasing due to an aging population. FIB4-T may be able to predict more accurate prognosis than JIS and ALBI-T in case of nephrotic syndrome etc., in cases where albumin value is lower than the patients without nephrotic syndrome or in case of constitutional jaundice. The advantage of the FIB4 index for these patients is that it does not use factors such as albumin, bilirubin, or prothrombin. The calculation requires only age and three laboratory test values (AST, ALT, and platelet count).

The recent development of nucleic acid analogues and direct acting antivirals has enabled the suppression of the HBV virus and effectively eradicated HCV. Thus, the probability of HCC in patients without liver inflammation will increase in the near future. Controlling viral replication would immediately improve and maintain liver function, whereas it takes a long time to improve fibrosis [27]. Although further studies are needed to provide proof of concept, in the future it may be possible to prescribe patient prognoses based on fibrosis rather than liver synthetic and metabolic ability.

This study has several limitations. All subjects examined were Japanese patients with HCC. Since the majority of Japanese patients are diagnosed with HCC at an early stage, compared with other countries, the usefulness of FIB4 might be limited to Japan. Furthermore, it is yet to be determined whether liver function or fibrosis is the better predictor of prognosis. In addition to hepatitis virus eradication, many new molecular target therapies for HCC will be implemented in the future and the conditions surrounding HCC are dramatically changing. Thus, the integrated scores of a new cohort will need to be determined and compared.

In conclusion, we have demonstrated that FIB4-T is a useful tool for prognostic prediction of HCC. Further examination is necessary to validate the usefulness of this score.

## 5. Conclusions

FIB4-T is useful in predicting the prognosis of patients with HCC from a new perspective.

## Figures and Tables

**Figure 1 cancers-11-00203-f001:**
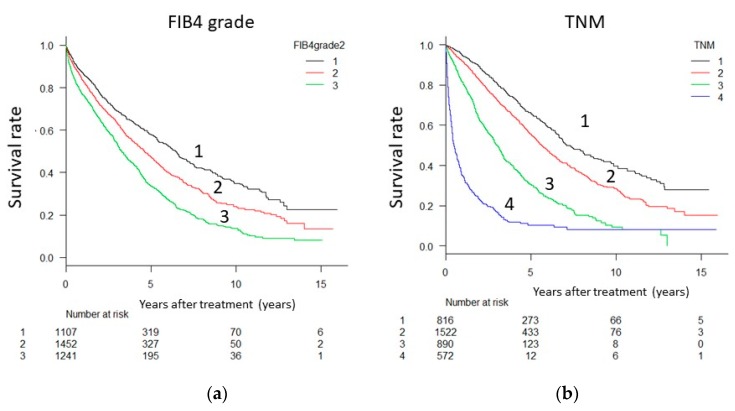
Kaplan–Meier curves according to (**a**) FIB4 grade and (**b**) Tumor, Node, Metastases (TNM) stage (data obtained from the Liver Cancer Study Group of Japan (n = 3800)). Statistically significant differences were observed among survival curves on the basis of FIB4 grade and TNM stage (log-rank test, *p* < 0.0001).

**Figure 2 cancers-11-00203-f002:**
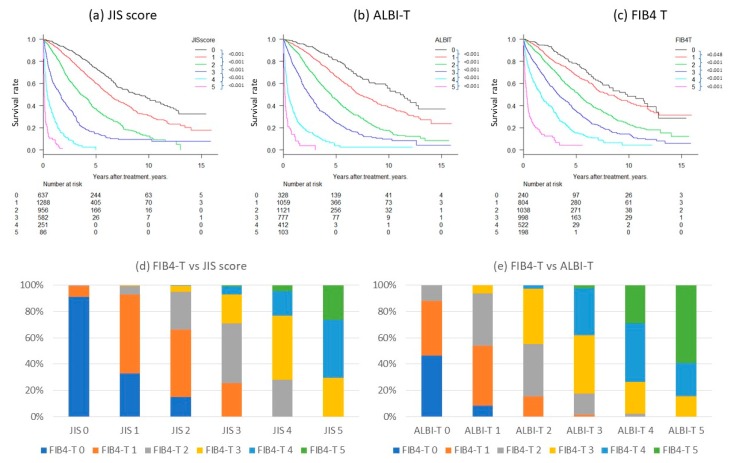
Overall survival rate according to (**a**) JIS score, (**b**) ALBI-T, and (**c**) FIB4-T (n = 3800) and Distribution of FIB4-T in (**d**) JIS score and (**e**) ALBI-T. The difference in survival between FIB4-T (0) and (1) was small compared with those in JIS score and ALBI-T, but prognosis of FIB4-T (0) was significantly better than prognosis of FIB4-T (1) (*p* = 0.048). The *p*-value for each group was FIB 4-T 0–1: 0.048, 1–2: <0.001, 2–3: <0.001, 3–4: <0.001, and 4–5: <0.001 (log-rank test). All *p*-values for each group in JIS score, ALBI-T were <0.001 (log-rank test). As prognostic models, good stratification was possible for all three systems (*p* < 0.0001). Distribution of FIB4-T in JIS score and ALBI-T were as shown in (**d**) and (**e**).

**Figure 3 cancers-11-00203-f003:**
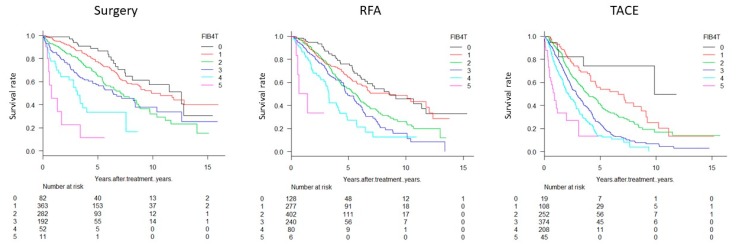
Survival curves stratified with FIB4-T for different treatments. Stratification was possible regardless of treatment type (surgery, *p* < 0.0001; RFA, *p* < 0.0001; TACE, *p* < 0.0001). Abbreviations: RFA, radiofrequency ablation; TACE, transcatheter arterial chemoembolization.

**Figure 4 cancers-11-00203-f004:**
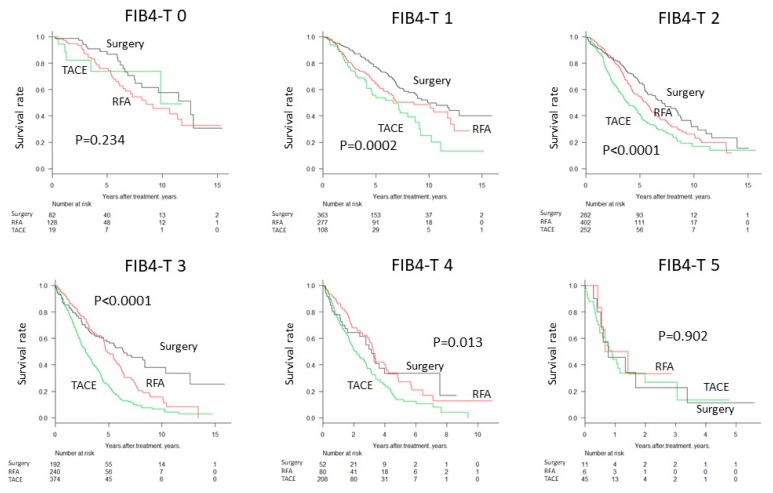
Survival curve stratified by treatment type (Surgery, RFA and TACE) for each Figure 4. grade. Abbreviations: RFA, radiofrequency ablation; TACE, transcatheter arterial chemoembolization.

**Table 1 cancers-11-00203-t001:** Definition of the Fibrosis-4 (FIB4)-T scoring system.

		FIB4 Grade		
		1	2	3
TNM stage by LCSGJ	I	FIB4-T 0	FIB4-T 1	FIB4-T 2
	II	FIB4-T 1	FIB4-T 2	FIB4-T 3
	III	FIB4-T 2	FIB4-T 3	FIB4-T 4
	IV	FIB4-T 3	FIB4-T 4	FIB4-T 5

Abbreviation: LCSGJ, Liver Cancer Study Group of Japan.

**Table 2 cancers-11-00203-t002:** Characteristics of the 3800 patients included in the study.

Variable	Value	%
Age, year (median (range)	70 (21–98)	
Sex (male)	2779	73.1
Cause of parenchymal disorder		
HBV	421	11.1
HCV	2566	67.5
Non-B, Non-C	777	20.4
Child–Pugh stage		
A	2660	70.0
B	918	24.2
C	222	5.8
FIB4 grade (FIB4 index), n		
1 (–3.25)	1107	29.1
2 (3.26–6.70)	1452	38.2
3 (6.71–)	1241	32.7
TNM stage by LCSGJ		
I	816	21.5
II	1522	40.1
III	890	23.4
IV	572	15.1
Initial treatment modality		
Surgery	982	25.8
Percutaneous ablation therapy	1201	31.6
PEIT	68	1.8
RFA	1133	29.8
TACE	1006	26.5
Others	237	6.2
BSC	474	12.5

Abbreviations: LCSGJ, Liver Cancer Study Group of Japan; PEIT, percutaneous ethanol injection therapy; RFA, radiofrequency ablation; TACE, transcatheter arterial chemoembolization; BSC, best supportive care.

**Table 3 cancers-11-00203-t003:** Overall survival of HCC patients stratified by FIB4-T, FIB4 grade, and TNM.

Factor		Patients Number	Survival Rate (%)			MST, Year
			1 Year	3 Years	5 Years	
FIB4-T	0	240	97	88	77	9.6
	1	804	94	80	66	8.0
	2	1038	92	72	51	5.2
	3	998	78	55	35	3.4
	4	522	60	36	14	1.8
	5	198	20	10	5	0.3
FIB4 grade	1	1107	88	69	59	6.8
	2	1452	84	62	48	4.6
	3	1241	77	52	32	3.2
TNM stage by LCSGJ	I	816	95	80	65	7.2
	II	1522	91	72	54	5.4
	III	890	81	49	29	2.8
	IV	572	33	17	9	0.4

Abbreviations: MST, median survival time; LCSGJ, Liver Cancer Study Group of Japan.

**Table 4 cancers-11-00203-t004:** Evaluation of scores (n = 3800).

Integrated Score	Likelihood Ratio χ^2^	Akaike Information Criterion
FIB4-T	765	26,460
JIS score	1208	26,018
ALBI-T	1168	26,058

Note: The Akaike information criterion (AIC) (Akaike, 1974) is a fined technique based on in-sample fit to estimate the likelihood of a model to predict/estimate the future values. A good model is the one that has minimum AIC among all the other models. The AIC can be used to select between the additive and multiplicative Holt–Winters models.

**Table 5 cancers-11-00203-t005:** Prognostic values defined by the Cox proportional hazard model.

Variables		Relative Risk	95%CI	*p*-Value
FIB4-T	0→1	1.303	1.002–1.695	0.048
	1→2	1.679	1.445–1.951	<0.001
	2→3	1.645	1.455–1.861	<0.001
	3→4	1.804	1.573–2.069	<0.001
	4→5	2.791	2.265–3.440	<0.001
JIS score	0→1	1.579	1.340–1.860	<0.001
	1→2	1.968	1.738–2.227	<0.001
	2→3	2.143	1.868–2.458	<0.001
	3→4	2.573	2.141–3.092	<0.001
	4→5	2.583	1.955–3.411	<0.001
ALBI-T	0→1	1.667	1.315–2.113	<0.001
	1→2	1.909	1.673–2.177	<0.001
	2→3	1.865	1.646–2.113	<0.001
	3→4	3.039	2.607–3.543	<0.001
	4→5	2.377	1.846–3.060	<0.001
AFP	>400 ng/mL	2.499	2.223–2.808	<0.001
PIVKA-II	>200 mAU/mL	2.588	2.338–2.865	<0.001
Portal invasion	positive	5.384	4.277–6.776	<0.001

Abbreviations: AFP, alpha-fetoprotein; PIVKA-II, protein induced by vitamin K absence; CI, confidence interval.

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
