# Peer review of "Utility of FIB4-T as a Prognostic Factor for Hepatocellular Carcinoma"

_cancers, 2019, doi:10.3390/cancers11020203_

Round 1
Reviewer 1 Report
Conflict of interest: none
The following report is divided into two parts containing:
A-General comments concerning the global appreciation of the work.
B-Specific comments concerning particular aspects of the article.
A-GENERAL ANALYSIS AND COMMENTS
The authors present an interesting and well-presented manuscript. The aim is clear and the results are convincing. A very large cohort of HCC patients with a long follow-up is studied. Three different score/indexes are used and compared in regard to their capacity to segregate patients’ survival. The new index that they introduce “FIB4-T” seems useful although comparable to ALBI-T and JIS scores.
However some modifications and improvements are needed to improve the overall quality of the study.
B-SPECIFIC ANALYSIS AND COMMENTS
Major comments (5)
Comment 1/5
Lines 157-160 and lines 167-8
The reader might wonder what does the p-values refer to ? since there are multiple models and that authors refer to stratifications, it is a bit confusing if the p-values refer to comparisons between models for similar stratifications? or if it relate to comparisons for each model referring to at least two significantly different stratification? … This should be clarified.
Comment 2/5
Figure 2, since the three models give similar survival curves, one can wonder whether or not the different scores and stratifications contain the same patients, meaning that the different scores would identify the same patients with different ways. It could be very informative for scientists and clinicians to provide Venn diagrams of all patients comparing all stratifications and scores such as:
diagram 1 including JIS, ALBI-T, FIB4-T with score :0
diagram 2 including JIS, ALBI-T, FIB4-T with score :1
diagram 3 including JIS, ALBI-T, FIB4-T with score :2
diagram 4 including JIS, ALBI-T, FIB4-T with score :3
diagram 5 including JIS, ALBI-T, FIB4-T with score :4
diagram 6 including JIS, ALBI-T, FIB4-T with score :5
Comment 3/5
Figure 3. Interesting to be able to see that the stratifications still remain with the different treatment alternatives,
One question that might be interesting to answer with this dataset is whether or not patients with a given FIB4-T score have different survival depending on the treatments ?
6 news graphics could be included with:
comparing survival of “FIB4-T=0” between “surgery”, “RFA” and “TACE”
comparing survival of “FIB4-T=1” between “surgery”, “RFA” and “TACE”
comparing survival of “FIB4-T=2” between “surgery”, “RFA” and “TACE”
comparing survival of “FIB4-T=3” between “surgery”, “RFA” and “TACE”
comparing survival of “FIB4-T=4” between “surgery”, “RFA” and “TACE”
comparing survival of “FIB4-T=5” between “surgery”, “RFA” and “TACE”
Comment 4/5
In Table 4 authors used “The Akaike information criterion (AIC)” that they defined as “an estimator of the relative quality of model statistical models”.
This statement is a bit misleading and confusing since there are many ways of evaluating a model. Authors used Cox model, which is usually not evaluated with this type of score. They could refer to significance of the overall model with p-values. However since here the different stratifications are of particular interest, the quality of the model might also be defined by “how well the stratifications identify patients with distinct survival”, which might require an assessment for every comparison, for example: “FIB4-T=0” vs “FIB4-T=1”, then “FIB4-T=1” vs “FIB4-T=2”, then FIB4-T=0 vs “FIB4-T=2”, …
Authors might provide further comments and explanations on the AIC score as well as additional quality estimation criteria.
Comment 5/5
Table 5 displays interesting data, however since the manuscript aims to compare FIB4 to JIS, ALBI-T, the authors might include these scores too in the same analysis so that the reader can have the possibility to do a side-by-side comparison.
Minor comments (2)
Comment 1/2
Line 63
Authors define liver fibrosis only as “epigenetic factor” which is a bit misleading. It might not be strictly biologically correct, this might be more referred as a much more complex dynamic biological process including multiple cell types and soluble mediators over a long period of time. This could be modified.
Comment 2/2
The resolution and quality of the figures might be improved.
Author Response
Major comments (5)
Comment 1/5
Lines 157-160 and lines 167-8
The reader might wonder what does the p-values refer to ? since there are multiple models and that authors refer to stratifications, it is a bit confusing if the p-values refer to comparisons between models for similar stratifications? or if it relate to comparisons for each model referring to at least two significantly different stratification? … This should be clarified.
The differences described in the legend of Figure 1(line157-160) were that between the curves in each figure. To make it clear, we change the description from “…on the bases of FIB4 grade and TNM stage (log-rank test, p<0.0001)” to “…of FIB4 grade (log-rank test, p<0.0001) and TNM stage (log-rank test, p<0.0001)” We also change the sentence at line 167-8 from “…, a significant difference was observed (p=0.048)” to “survival of FIB4-T (0) was significantly longer than that of FIB4-T (1)(p=0.048). The P-value between other combinations of FIB4T grades were all less than 0.001.”
Comment 2/5
Figure 2, since the three models give similar survival curves, one can wonder whether or not the different scores and stratifications contain the same patients, meaning that the different scores would identify the same patients with different ways. It could be very informative for scientists and clinicians to provide Venn diagrams of all patients comparing all stratifications and scores such as:
diagram 1 including JIS, ALBI-T, FIB4-T with score :0
diagram 2 including JIS, ALBI-T, FIB4-T with score :1
diagram 3 including JIS, ALBI-T, FIB4-T with score :2
diagram 4 including JIS, ALBI-T, FIB4-T with score :3
diagram 5 including JIS, ALBI-T, FIB4-T with score :4
diagram 6 including JIS, ALBI-T, FIB4-T with score :5
: Since it was difficult to compare 3 models simultaneously with Venn diagram, we made two bar graphs that indicated the difference between FIB4-T and JIS score, and between FIB4-T and ALBI-T and showed as Figure 2d and 2e. . We added a sentence in the text (line xx) and in Figure legend to explain the figures. The added sentence was “Distributions of the patients in different indices were shown in Figure 2d and 2e.”
Comment 3/5
Figure 3. Interesting to be able to see that the stratifications still remain with the different treatment alternatives,
One question that might be interesting to answer with this dataset is whether or not patients with a given FIB4-T score have different survival depending on the treatments ?,
6 news graphics could be included with:
comparing survival of “FIB4-T=0” between “surgery”, “RFA” and “TACE”
comparing survival of “FIB4-T=1” between “surgery”, “RFA” and “TACE”
comparing survival of “FIB4-T=2” between “surgery”, “RFA” and “TACE”
comparing survival of “FIB4-T=3” between “surgery”, “RFA” and “TACE”
comparing survival of “FIB4-T=4” between “surgery”, “RFA” and “TACE”
comparing survival of “FIB4-T=5” between “surgery”, “RFA” and “TACE”
: We made new figures (Fig. 4) in order to compare the prognosis of Surgery, RFA and TACE in each FIB4-T grade. We added a sentence in the text (line 153-154) and Figure legend. The sentence is “Survival curves of the patients stratified by treatment types (Surgery, RFA and TACE) in each FIB4-T grade were shown in Figure 4.”
Comment 4/5
In Table 4 authors used “The Akaike information criterion (AIC)” that they defined as “an estimator of the relative quality of model statistical models”.
This statement is a bit misleading and confusing since there are many ways of evaluating a model. Authors used Cox model, which is usually not evaluated with this type of score. They could refer to significance of the overall model with p-values. However since here the different stratifications are of particular interest, the quality of the model might also be defined by “how well the stratifications identify patients with distinct survival”, which might require an assessment for every comparison, for example: “FIB4-T=0” vs “FIB4-T=1”, then “FIB4-T=1” vs “FIB4-T=2”, then FIB4-T=0 vs “FIB4-T=2”, …
Authors might provide further comments and explanations on the AIC score as well as additional quality estimation criteria.
:According to the reviewer’s suggestion, we revised an explanation of AIC (line xx) and also showed the results of the comparison between different scores in text (line 207-211) and in the legend of Figure 2 as we explained in the answer to the comment 1/5. The revised sentences at line 207-211 were “Akaike information criterion (AIC) (Akaike, 1974) is a fined technique based on in-sample fit to estimate the likelihood of a model to predict/estimate the future values. A good model is the one that has minimum AIC among all the other models”.
Comment 5/5
Table 5 displays interesting data, however since the manuscript aims to compare FIB4 to JIS, ALBI-T, the authors might include these scores too in the same analysis so that the reader can have the possibility to do a side-by-side comparison.
: According to the reviewer’s comment, we added the data of JIS score and ALBI-T in Table 5.
Minor comments (2)
Comment 1/2
Line 63
Authors define liver fibrosis only as “epigenetic factor” which is a bit misleading. It might not be strictly biologically correct, this might be more referred as a much more complex dynamic biological process including multiple cell types and soluble mediators over a long period of time. This could be modified.
: According to the reviewer’s suggestion, we change the sentence of line 63 from “Liver fibrosis is one of epigenetic factors closely related to carcinogenesis” to “Liver fibrosis progresses by various factors such as genetic factors, viral or non-viral inflammation and drugs, and closely related to carcinogenesis”.
Comment 2/2
The resolution and quality of the figures might be improved.
: Thank you for your suggestion. We revised all figures.
Reviewer 2 Report
I have read your article about anew staging system for prognosis in HCC patients. Authors clarified the equality of FIB4-T compared with JIS and ALBI-T. However, various etiological HCC patients are rolled in this study, many hepatologists and hepatobiliary surgeons may have interests in the prognosis in every major etiologies (HBV, HCV, others), RFS or DFS in patients with curative intervention.
Author Response
I have read your article about anew staging system for prognosis in HCC patients. Authors clarified the equality of FIB4-T compared with JIS and ALBI-T. However, various etiological HCC patients are rolled in this study, many hepatologists and hepatobiliary surgeons may have interests in the prognosis in every major etiologies (HBV, HCV, others), RFS or DFS in patients with curative intervention.
: Thank you for your advices. As you said, many hepatologists and hepatobiliary surgeons may have interests in the prognosis in every major etiologies (HBV, HCV, others), RFS or DFS in patients with curative intervention. However, the main topic of this study was not to show the differences of RFS or OS in different etiologies but was to clarify ability of FIB4T so that we dare not to show the results.
Reviewer 3 Report
Kariyama et al. set out to determine if newly created FIB4-T scoring system could predict prognosis in patients with HCC.
FIB4-T represents a combination of FIB4 metric (age, transaminases, and platelet count) previously used to assess hepatic fibrosis and standard TNM. 3,800 patients were enrolled to evaluate FIB4-T as a prognosticator.
In comparison to FIB4 and TNM alone, FIB4T clearly allowed further stratification, and risk of mortality progressively increased with each FIB4-T step. Comparison with established JIS and ALBI-T scores demonstrated that FIB4-T was a reliable metric. At the same time, it was shown that FIB4-T performed slightly worse than both JIS and ALBI-T.
There are multiple scoring systems to predict prognosis in patients with HCC: TNM, Child-Pugh, Okuda, Barcelona staging classification, and ALBI-T. Major benefits of proposed FIB4-T is that it is fully objective and requires only basic labs (complete blood count (CBC) and comprehensive metabolic panel (CMP)). This may make it more attractive than, for example, Child-Pugh scoring system which partly relies on subjective factors (such as ascites and encephalopathy). Likewise, it may potentially be more accurate than JIS which is a combination of Child-Pugh and LCSGJ. However, it is hard to understand why FIB4-T may be superior to ALBI-T which also relies on routinely measured albumin and bilirubin and which performed better despite listed limitations. From the economic standpoint, ALBI-T and FIB4-T do not appear to be different since both require only basic lab tests (CBC and CMP) routinely done in all patients.
Major comment
The clinical significance does not seem to be high. This should be better addressed/balanced. As of now, FIB4-T appears to be an alternative scoring system which performs slightly worse than ALBI-T. It would be better if, for example, this scoring system benefited some subpopulation.
Minor comments
1). It is a little confusing to read in Abstract and Results that the mean observation period was 37 months while 5-, 10-, and 15-year survival was assessed. Does it mean that less patients were evaluated at 10- and 15-year time-points? If so, authors should indicate how many patients were included in each time-point.
2). Did monitoring differ in patients with different FIB4-T scores (i.e. were patients with smaller FIB4-T scores less frequently monitored which could result in unrecorded deaths)?
3). The authors may want to clarify whether they used c-JIS or bm-JIS.
Author Response
Major comment
The clinical significance does not seem to be high. This should be better addressed/balanced. As of now, FIB4-T appears to be an alternative scoring system which performs slightly worse than ALBI-T. It would be better if, for example, this scoring system benefited some subpopulation.
: The reviewer recommended to explain the benefit of FIB4T. The advantage of the FIB4T is that it does not use factors such as albumin, bilirubin, or prothrombin so that we could predict the prognosis more accurately in patients with nephrotic syndrome and with frequently observed constitutional jaundice. We have already described this explanation in Discussion (line 251-255)
Minor comments
1). It is a little confusing to read in Abstract and Results that the mean observation period was 37 months while 5-, 10-, and 15-year survival was assessed. Does it mean that less patients were evaluated at 10- and 15-year time-points? If so, authors should indicate how many patients were included in each time-point.
: We have already shown the information as the number of patients at risk in Fig. 1, 2, and 3.
2). Did monitoring differ in patients with different FIB4-T scores (i.e. were patients with smaller FIB4-T scores less frequently monitored which could result in unrecorded deaths)?
: Since all of them suffered from HCC, we follow up all patients every 3~4 months and the monitoring interval was not different between low FIB4-T group and high FIB4-T group.
3). The authors may want to clarify whether they used c-JIS or bm-JIS.
: We used c-JIS score and have explained by adding a reference (ref. 5).
Round 2
Reviewer 1 Report
Dear colleagues,
I feel that the authors responded here adequately to the suggested comments and that the quality of the manuscript has been improved, in order to provide to the readers more complete and detailed informations,
The informations provided by this study are interesting for both clinicians and more basic researchers,
Best,